# JointMatch: A Unified Approach for Diverse and Collaborative Pseudo-Labeling to Semi-Supervised Text Classification

**Henry Peng Zou**  **Cornelia Caragea**
Computer Science
University of Illinois Chicago
pzou3@uic.edu  cornelia@uic.edu

## Abstract

Semi-supervised text classification (SSTC) has gained increasing attention due to its ability to leverage unlabeled data. However, existing approaches based on pseudo-labeling suffer from the issues of pseudo-label bias and error accumulation. In this paper, we propose JointMatch, a holistic approach for SSTC that addresses these challenges by unifying ideas from recent semi-supervised learning and the task of learning with noise. JointMatch adaptively adjusts classwise thresholds based on the learning status of different classes to mitigate model bias towards current easy classes. Additionally, JointMatch alleviates error accumulation by utilizing two differently initialized networks to teach each other in a cross-labeling manner. To maintain divergence between the two networks for mutual learning, we introduce a strategy that weighs more disagreement data while also allowing the utilization of high-quality agreement data for training. Experimental results on benchmark datasets demonstrate the superior performance of Joint-Match, achieving a significant 5.13% improvement on average. Notably, JointMatch delivers impressive results even in the extremely-scarce-label setting, obtaining 86% accuracy on AG News with only 5 labels per class. We make our code available at https://github.com/HenryPengZou/JointMatch.

## 1 Introduction

The success of deep learning models often heavily depends on the availability of large amounts of labeled data (He et al., 2016; Vaswani et al., 2017). However, the labeled data for many tasks are often expensive, difficult, and time-consuming to obtain. By contrast, acquiring unlabeled data is more cost-effective and convenient in many scenarios. This has led to a surge of interest in semi-supervised learning, which aims to enhance learning performance with limited labeled samples by leveraging large amounts of unlabeled data (Berthelot et al., 2019; Sohn et al., 2020).

Recently, the combination of pseudo-labeling and consistency regularization has become a popular paradigm for semi-supervised learning (Sohn et al., 2020; Zhang et al., 2021; Sosea and Caragea, 2022). Pseudo-labeling (Lee et al., 2013) uses a fixed threshold to select the model's high-confidence predictions as pseudo-labels for further training, whereas consistency regularization (Sajjadi et al., 2016) enforces the model to make similar predictions for perturbed versions of the same data. For example, UDA (Xie et al., 2020) applies strong data augmentations, such as back-translation, to unlabeled data and minimizes the divergence between model predictions of input and its augmented views. FixMatch (Sohn et al., 2020) uses the pseudo-label generated from weakly-augmented unlabeled data to supervise the strongly augmented version of the same data. SAT (Chen et al., 2022) improves FixMatch by training a meta-learner to re-rank different augmentations based on their similarities with the original data. These methods require a pre-defined high-confidence threshold to generate high-quality pseudo-labels for good performance. However, there are some potential limitations: (1) Setting a fixed threshold for pseudo-label selection neglects the varied difficulties of learning different classes (Zhang et al., 2021; Wang et al., 2023b,a). This can cause the model bias toward easy classes, as more pseudo-labels will be generated for current easy classes (see Figure 1a); and (2) If these pseudo-labels are incorrect and used to train the model, the model can be worse and produce more inaccurate pseudo-labels, progressively accumulating its error and degenerating its performance (see Figure 1c) (Arazo et al., 2020).

To address these issues, we propose Joint-Match, a diverse and collaborative pseudo-labeling approach for semi-supervised text classification (SSTC). JointMatch is a holistic framework that

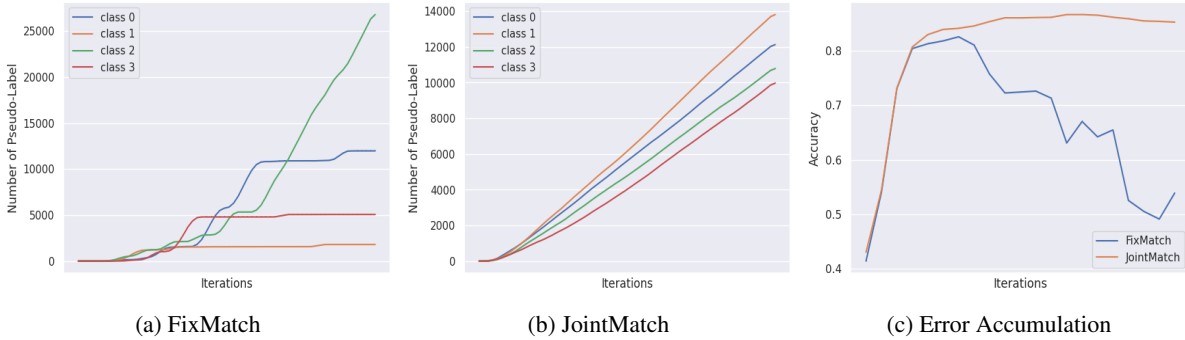

| (a) FixMatch | (b) JointMatch | (c) Error Accumulation |

Figure 1: Number of pseudo labels (a, b) and validation accuracy (c) of FixMatch and JointMatch on AG News with 10 labels per category. Pseudo-labeling approaches that use a fixed threshold, such as FixMatch, suffer from pseudo-label bias towards easy classes and error accumulation along the training. In contrast, JointMatch effectively balances pseudo-labels and avoids degenerated performance caused by error accumulation.

unifies ideas from recent semi-supervised learning and the task of learning with noise for SSTC. Inspired by FlexMatch (Zhang et al., 2021) and FreeMatch (Wang et al., 2023b), we adaptively adjust classwise thresholds based on the estimated learning status for different classes at different times. This enables difficult classes at the current iteration to have lower local thresholds, thereby facilitating more pseudo-labels to be produced and used for learning these classes (see Figure 1b). Following the idea of co-training (Blum and Mitchell, 1998), JointMatch simultaneously trains two differently initialized networks and uses them to teach each other in a cross-labeling manner to alleviate error accumulation, as different networks can filter different types of noise.

Nevertheless, with the increase of training iterations, those networks will slowly converge and reduce to one network and thus will again suffer from the issue of error accumulation. Inspired by Co-Teaching+ (Yu et al., 2019) and Decoupling (Malach and Shalev-Shwartz, 2017), we propose to give more weight to disagreement data to keep the two networks diverged while also allowing the utilization of high-quality agreement data for training. The relationship and difference between our JointMatch and related techniques are discussed in detail in Section 2.5.

We evaluate the proposed JointMatch on three commonly studied SSTC benchmark datasets. Experimental results indicate the superior performance of JointMatch, obtaining a significant 5.13% average improvement over the latest work SAT. We also analyze the performance of JointMatch with varying numbers of labeled data. The results show that JointMatch can deliver impressive results even

in the extremely-scarce-labels setting, achieving 86% accuracy on AG News with only 5 labels per class. We provide comprehensive ablation studies and analysis to understand each part of JointMatch.

## 2 JointMatch

### 2.1 Overview

In this section, we introduce our proposed JointMatch for semi-supervised text classification. JointMatch is a unified approach that integrates ideas and components from recent semi-supervised learning and the task of learning with noise. Specifically, we utilize three key techniques, i.e, (i) adaptive local thresholding; (ii) cross-labeling; (iii) weighted disagreement & agreement update, to address the limitations we have identified: (a) bias towards easy classes; (b) error accumulation; (c) tradeoff between divergence & consensus.

The main pipeline for JointMatch is shown in Figure 2. For each batch of unlabeled data $\mathcal{U}$, we first apply both weak data augmentation $\alpha(\cdot)$ and strong data augmentation $A(\cdot)$, such as synonym replacement and back translation, respectively. The weakly augmented data are then forwarded to two differently initialized models $f$ and $g$ to make predictions. High-confidence predictions that pass the adaptive local threshold $\tau_t(c)$ are selected as pseudo-labels. Especially, the pseudo-labels generated by one model are used to teach its peer network, i.e., cross-labeling. The unlabeled loss $\mathcal{L}_u$ is computed between generated pseudo-labels and model predictions of strongly augmented data. Here, we weigh more disagreement data (where both networks have different label predictions) to keep the two networks diverged but also allow the utilization of agreement data that are more likely

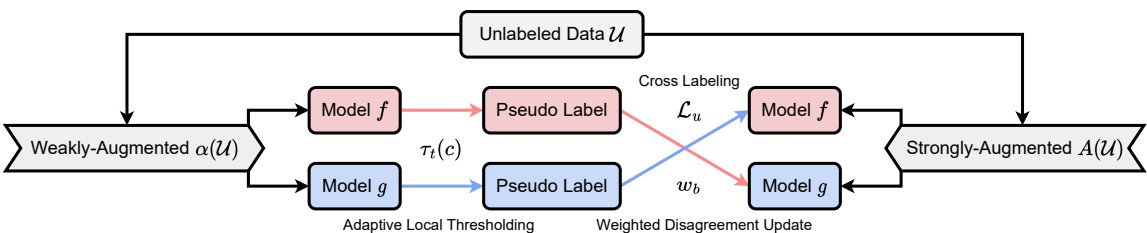

Figure 2: Pipeline of JointMatch. Unlabeled data undergo both weak and strong data augmentation. Weakly augmented data are first fed into two differently initialized models. Predictions with confidence surpassing adaptive local thresholds are selected as pseudo labels for training. Notably, pseudo labels from one model are used to teach its peer network, i.e, cross-labeling. Unlabeled data loss is then computed between pseudo labels from weakly-augmented data and model predictions from strongly augmented data. Here, we weigh more disagreement data to keep the two models diverged for effective mutual learning.

to be correct. A complete algorithm for JointMatch is presented in Algorithm 1. Next, we explain our key components in detail.

## 2.2 Adaptive Local Thresholding

Pseudo-labeling-based semi-supervised text classification algorithms use a *fixed* threshold to select high-confidence unlabeled data as pseudo-labels for training. However, this ignores the varied difficulties of learning different classes at different time steps, and may result in that easier classes can generate more pseudo-labels than hard classes for learning and can cause increasing model bias along training (see Figure 1a). Inspired by Flex-Match (Zhang et al., 2021) and FreeMatch (Wang et al., 2023b), instead of using a fixed threshold, our JointMatch estimates the learning status of different classes and adaptively adjusts the local thresholds for different classes to produce more diversified pseudo-labels (see Figure 1b). Specifically, at each time step $t$, we first estimate the classwise learning status $\tilde{p}_t = [\tilde{p}_t(1), \tilde{p}_t(2), ..., \tilde{p}_t(C)]$ via the *exponential moving average* of model's predicted probability on unlabeled data:

$$\tilde{p}_t = \lambda\tilde{p}_{t-1} + (1-\lambda)\frac{1}{\mu B}\sum_{b=1}^{\mu B} q_b \qquad (1)$$

where $\tilde{p}_0 = [1/C, 1/C, ..., 1/C]$, $C$ is the number of classes, $B$ is the batch size of labeled data, $\mu$ is the ratio of unlabeled data to labeled data, $\lambda \in (0,1)$ is the momentum parameter, $q_b = p_m(\alpha(u_b))$ denotes model's predicted class distribution on weak-augmented unlabeled data. Then we normalize the estimated learning status and adjust the local threshold for each class $c$ from the pre-defined threshold $\tau$:

$$\tau_t(c) = \text{MaxNorm}(\tilde{p}_t(c)) \cdot \tau = \frac{\tilde{p}_t(c)}{\max_c(\tilde{p}_t(c))} \cdot \tau \qquad (2)$$

By doing this, difficult classes at the current iteration will have lower local thresholds, encouraging more pseudo-labels to be generated and utilized for training these classes.

## 2.3 Cross-Labeling

Another issue of pseudo-labeling, or more generally, self-training, is error accumulation: If the generated predictions are incorrect and the model is trained on them, the model can become worse and worse, continually producing more noisy pseudo-labels and accumulating its error (see Figure 1c). Inspired by Co-Training (Blum and Mitchell, 1998), our JointMatch involves the simultaneous training of two networks with different initializations. These networks are utilized to mutually instruct each other through *cross-labeling*. This strategy mitigates error accumulation because different networks can filter out different noises.

More formally, given two differently initialized networks $f$ and $g$, each network first selects its own high-confidence predictions of unlabeled data as pseudo-labels for the other network. The selected pseudo-labels are then used to compute unlabeled data loss $\mathcal{L}_u$ to update the parameters for its peer network:

$$\mathcal{L}_u^f = \frac{1}{\mu B}\sum_{b=1}^{\mu B} \mathbb{1}(\max(q_b^g) \geq$$
$$\tau_t^g(\arg\max(q_b^g)))\mathcal{H}(\hat{q}_b^g, Q_b^f)$$

$$\mathcal{L}_u^g = \frac{1}{\mu B}\sum_{b=1}^{\mu B} \mathbb{1}(\max(q_b^f) \geq$$
$$\tau_t^f(\arg\max(q_b^f)))\mathcal{H}(\hat{q}_b^f, Q_b^g)$$
$$(3)$$

**Algorithm 1** JointMatch algorithm.

---

1: **Input:** Differently initialized model $f$, $g$, number of classes $C$, labeled batch $\mathcal{X} = \{(x_b, y_b) : b \in (1, 2, \ldots, B)\}$, unlabeled batch $\mathcal{U} = \{u_b : b \in (1, 2, \ldots, \mu B)\}$, where $B$ is the batch size of labeled data, $\mu$ is the ratio of unlabeled data to labeled data, unsupervised loss weight $w_u$, EMA decay $\lambda$, fixed threshold $\tau$, disagreement weight $\delta$, weak and strong data augmentations $\alpha(\cdot)$, $A(\cdot)$.

2: **for** model $f$, $g$ **do**

3:     $\mathcal{L}_s = \frac{1}{B} \sum_{b=1}^{B} \mathcal{H}(y_b, p_m(\alpha(x_b)))$ {*Compute loss for labeled data, $p_m(\cdot)$ denotes for model prediction*}

4:     $\tilde{p}_t = \lambda \tilde{p}_{t-1} + (1-\lambda)\frac{1}{\mu B} \sum_{b=1}^{\mu B} q_b$ {*Estimate learning status for different classes, $q_b$ is an abbreviation of $p_m(\alpha(u_b))$*}

5:     **for** $c = 1$ to $C$ **do**

6:         $\tau_t(c) = \text{MaxNorm}(\tilde{p}_t(c)) \cdot \tau$ {*Adjust the local threshold for each class based on its learning status*}

7:     **end for**

8: **end for**

9: **for** $b = 1$ to $\mu B$ **do**

10:     $w_b = \delta \mathbb{1}(\hat{q}_b^f \neq \hat{q}_b^g) + (1-\delta)\mathbb{1}(\hat{q}_b^f = \hat{q}_b^g)$ {*Compute sample weight based on prediction disagreement, $\hat{q}_b$ denotes the generated hard label*}

11: **end for**

12: $\mathcal{L}_u^f = \frac{1}{\mu B} \sum_{b=1}^{\mu B} w_b \mathbb{1}\left(\max\left(q_b^g\right) \geq \tau_t^g(\arg\max\left(q_b^g\right))\right) \mathcal{H}(\hat{q}_b^g, Q_b^f)$

    $\mathcal{L}_u^g = \frac{1}{\mu B} \sum_{b=1}^{\mu B} w_b \mathbb{1}\left(\max\left(q_b^f\right) \geq \tau_t^f(\arg\max\left(q_b^f\right))\right) \mathcal{H}(\hat{q}_b^f, Q_b^g)$ {*Compute loss for unlabeled data, the pseudo-label generated by one network is used to supervise another network, $Q_b$ is an abbreviation of $p_m(A(u_b))$*}

13: **Return:** $\mathcal{L}_s^f + w_u \mathcal{L}_u^f$,   $\mathcal{L}_s^g + w_u \mathcal{L}_u^g$

---

where $q_b$ and $Q_b$ denote model's predicted class distributions on weakly and strongly augmented data, respectively, $\hat{q}_b$ is the one-hot label that is converted from $q_b$, and $\mathcal{H}$ refers to cross-entropy loss. Compared to self-training, this approach displays a zigzag-shaped error flow, which helps to avoid direct error accumulation within a single network.

## 2.4 Weighted Disagreement & Agreement Update

Two differently initialized networks can have varied learning abilities to filter different types of error in the initial training stage, allowing them to learn from its peer. However, as the training goes on, those networks will gradually converge, and the co-training approach will degenerate to self-training and thus again suffer from error accumulation. To address this problem, Decoupling (Malach and Shalev-Shwartz, 2017) and Co-Teaching+ (Yu et al., 2019) propose to update networks only by data with disagreed network predictions and find that this strategy is effective in keeping two networks diverged. Nevertheless, their approaches completely ignore the agreement data, where both networks have the same prediction. We argue that those agreement data are a valuable learning signal, as they are more likely to receive correct pseudo-

labels and should also be utilized. To this end, we propose to compute a loss weight $w_b$ for each unlabeled data $u_b$ that weighs more disagreement data to keep two networks diverged while also allowing the utilization of agreement data:

$$w_b = \delta \mathbb{1}(\hat{q}_b^f \neq \hat{q}_b^g) + (1-\delta)\mathbb{1}(\hat{q}_b^f = \hat{q}_b^g) \quad (4)$$

where $\delta \in (0.5, 1)$ is the disagreement weight. The unlabeled data loss $\mathcal{L}_u$ thus becomes:

$$\mathcal{L}_u^f = \frac{1}{\mu B} \sum_{b=1}^{\mu B} w_b \mathbb{1}(\max(q_b^g) \geq$$
$$\tau_t^g(\arg\max\left(q_b^g\right)))\mathcal{H}(\hat{q}_b^g, Q_b^f)$$
$$\mathcal{L}_u^g = \frac{1}{\mu B} \sum_{b=1}^{\mu B} w_b \mathbb{1}(\max(q_b^f) \geq$$
$$\tau_t^f(\arg\max(q_b^f)))\mathcal{H}(\hat{q}_b^f, Q_b^g)$$
$$(5)$$

Our experiment result in Section 4.2 shows that this approach is more effective than the method using (i) only disagreement data; (2) only agreement data; (3) weighting both kinds of data equally. Note that this method can be further improved by adaptively adjusting the disagreement weight based on the disagreement degree of two networks or based on

| | Semi-Supervised | | | Fully-Supervised | | SSTC |
|---|---|---|---|---|---|---|
| | FixMatch/ SAT | FlexMatch/ FreeMatch | Co-Training | Decoupling | Co-Teaching+ | JointMatch |
| Pseudo-Labeling | ✓ | ✓ | ✓ | ✗ | ✗ | ✓ |
| Weak-Strong Augmentation | ✓ | ✓ | ✗ | ✗ | ✗ | ✓ |
| **Adaptive Local Threshold** | ✗ | ✓ | ✗ | ✗ | ✗ | ✓ |
| Double Networks | ✗ | ✗ | ✓ | ✓ | ✓ | ✓ |
| **Cross Labeling** | ✗ | ✗ | ✓ | ✗ | ✓ | ✓ |
| Disagreement Update | ✗ | ✗ | ✗ | ✓ | ✓ | ✓ |
| **Weighted Disagree & Agree Update** | ✗ | ✗ | ✗ | ✗ | ✗ | ✓ |

Table 1: Comparison of closely related techniques with our JointMatch approach. In the first column, "Pseudo-Labeling": select high-confidence predictions as pseudo-labels for training; "Weak-Strong Augmentation": use the pseudo-labels generated by weakly-augmented sample to supervise the strongly-augmented sample; "Disagreement Update": update networks by only samples receives different pseudo-label from the two networks.

different time steps for curriculum learning. However, this topic is outside the focus and scope of this paper, and we leave it to interested researchers for future exploration.

The overall objective for training one network in JointMatch is:

$$\mathcal{L} = \mathcal{L}_s + w_u \mathcal{L}_u \qquad (6)$$

where $w_u$ represents the loss weight for unlabeled data loss $\mathcal{L}_u$ and the supervised loss $\mathcal{L}_s$ is given by:

$$\mathcal{L}_s = \frac{1}{B} \sum_{b=1}^{B} \mathcal{H}(y_b, p_m(\alpha(x_b))) \qquad (7)$$

## 2.5 Relation to Other Approaches

In this section, we compare our JointMatch algorithm with closely related approaches in Table 1. Our goal is to identify the connections among them and point out the key techniques that make JointMatch effective in semi-supervised text classification. FixMatch (Sohn et al., 2020) and SAT (Chen et al., 2022) employ the consistency regularization between weak and strong augmentation with pseudo-labeling to leverage unlabeled data. However, this approach uses a fixed threshold for pseudo-labeling, and often suffers from model bias towards easy class and error accumulation along their training. FlexMatch (Zhang et al., 2021) and FreeMatch (Wang et al., 2023b) propose to adaptively adjust local thresholds based on the estimated learning status of each class, which alleviates the bias towards easy classes. To address the issue of error accumulation, Co-Training (Blum and Mitchell,

1998) trains two networks simultaneously and supervises each network by the generated pseudo-labels from its peer network.

Decoupling (Malach and Shalev-Shwartz, 2017) and Co-Teaching+ (Yu et al., 2019) observe that update by disagreement data can make two networks in Co-Training diverged and thus more effective. However, when updating their network, they completely ignore agreement data, which is also more likely to receive correct pseudo-labels in semi-supervised learning. Therefore, we propose utilizing both disagreement and agreement data for training, but weighting disagreement data higher to keep the two networks diverged while also exploiting the high-quality pseudo-labels from agreement data.

Our JoinMatch framework unifies ideas from the approaches described above into a single, effective framework for semi-supervised text classification. The three key techniques in JointMatch are: (i) adaptive local thresholding; (ii) cross-labeling; and (iii) weighted disagreement & agreement update.

## 3 Experiments

### 3.1 Experimental Setup

**Datasets and Metrics.** We evaluate JointMatch on three standard semi-supervised text classification benchmarks: AG News (Zhang et al., 2015), Yahoo! Answers (Chang et al., 2008) and IMDB (Maas et al., 2011). Following (Chen et al., 2022, 2020; Li et al., 2021), we use the original test set and randomly sample from the training set to construct our training unlabeled set and validation set. Table 2 presents the dataset statistics and split information. We report the mean and standard

| Dataset | Label Type | # Classes | # Unlabeled | # Validation | # Test |
|---|---|---|---|---|---|
| AG News | News Topic | 4 | 5000 | 2000 | 1900 |
| Yahoo! Answer | AQ Topic | 10 | 5000 | 2000 | 6000 |
| IMDB | Review Sentiment | 2 | 5000 | 1000 | 12500 |

Table 2: Dataset statistics and split information. The number of unlabeled data, validation data and test data in the table means the number of data per class.

| | AG News (c=4) | | Yahoo! (c=10) | | IMDB (c=2) | | |
|---|---|---|---|---|---|---|---|
| Methods | Accuracy | Macro-F1 | Accuracy | Macro-F1 | Accuracy | Macro-F1 | Average |
| BERT | 69.18 ± 3.7 | 68.27 ± 3.5 | 58.11 ± 1.6 | 57.38 ± 1.9 | 63.16 ± 1.4 | 62.93 ± 1.6 | 63.17 |
| UDA | 76.69 ± 3.2 | 76.51 ± 3.0 | 59.32 ± 2.0 | 58.47 ± 2.3 | 64.88 ± 1.7 | 64.57 ± 1.5 | 66.74 |
| MixText | 78.07 ± 2.8 | 77.23 ± 3.5 | 59.93 ± 1.9 | 59.24 ± 1.8 | 65.22 ± 1.1 | 65.78 ± 1.2 | 67.58 |
| FixMatch | 80.22 ± 2.4 | 78.98 ± 2.1 | 60.17 ± 1.7 | 59.86 ± 1.5 | 64.52 ± 1.6 | 64.31 ± 1.4 | 68.18 |
| SAT | 85.43 ± 1.2 | 85.30 ± 1.5 | 61.33 ± 1.5 | 60.96 ± 1.4 | 68.96 ± 1.7 | 68.92 ± 1.6 | 71.82 |
| JointMatch (Ours) | **87.68 ± 0.5** | **87.64 ± 0.5** | **66.58 ± 0.7** | **66.09 ± 0.8** | **76.91 ± 4.5** | **76.79 ± 4.6** | **76.95** |

Table 3: Performance (accuracy (%) and macro-F1 (%)) comparison with baselines on different text classification datasets. JointMatch delivers best results on all benchmark datasets, surpassing the latest work SAT (Chen et al., 2022) on SSTC by 5.13% on average. The best results are shown in **blue**. $c$: number of classes.

deviation of accuracy and macro-F1 from five runs with different model parameter initialization.

**Baselines.** We compare JointMatch with several popular and recent approaches: UDA (Xie et al., 2020), MixText (Chen et al., 2020), FixMatch (Sohn et al., 2020) and SAT (Chen et al., 2022). We also compare to the vanilla ensemble of FixMatch and FreeMatch (Wang et al., 2023b) to further demonstrate the effectiveness of JointMatch.

**Implementation Details.** Following (Chen et al., 2022, 2020), we use the BERT-based-uncased model as our backbone model and the Hugging-Face Transformers (Wolf et al., 2020) library for the implementation. We adopt the same data augmentation techniques for fair comparisons, i.e., synonym replacement for weak augmentation and back-translation for strong augmentation, in all baselines. In detail, for back translation, we translate texts into German and then translate them back into English; for synonym replacement, we randomly substitute 30% of words with WordNet synonyms. A complete list of our hyper-parameters is provided in Appendix A, and our code is released.

## 3.2 Comparison with Baselines

We summarize the comparison with baselines on different text classification datasets in Table 3. Following SAT (Chen et al., 2022), we randomly sample $N_c$ samples per class as labeled data for train-

| #Labels/Class | 5 | 10 | 15 | 25 | 100 | 1000 |
|---|---|---|---|---|---|---|
| FixMatch | 56.13 | 59.81 | 65.07 | 63.85 | 68.07 | 68.81 |
| FixMatch-Ensemble | 57.08 | 61.62 | 66.52 | 65.92 | 68.40 | 70.42 |
| JointMatch | **64.08** | **65.52** | **67.51** | **67.24** | **69.74** | **71.39** |

Table 4: Accuracy on Yahoo! Answer with varying numbers of labeled data.

| #Labels/Class | 5 | 10 | 15 | 25 | 100 | 1000 |
|---|---|---|---|---|---|---|
| FixMatch | 70.29 | 80.25 | 82.32 | 85.96 | 87.79 | 89.57 |
| FixMatch-Ensemble | 72.01 | 80.86 | 84.14 | 85.33 | 86.86 | 88.61 |
| FreeMatch-Ensemble | 77.03 | 85.83 | 86.55 | 87.36 | 87.79 | 89.67 |
| JointMatch | **86.00** | **87.68** | **88.33** | **88.50** | **89.07** | **90.29** |

Table 5: Accuracy on AG News with varying numbers of labeled data. JointMatch delivers significant improvements over FixMatch, especially for extremely low-shot settings, indicating its effectiveness in leveraging unlabeled data.

| Dataset | Empathetic Dialogues | Go Emotions | Hurricane | Average Accuracy |
|---|---|---|---|---|
| # Classes | 32 | 27 | 8 | - |
| BERT (2019) | 21.63 | 28.86 | 71.72 | 40.74 |
| FixMatch (2020) | 24.40 | 30.06 | 73.44 | 42.63 |
| SAT (2022) | 29.25 | 30.36 | 74.06 | 44.56 |
| FreeMatch (2023) | 30.15 | 32.74 | 77.03 | 46.64 |
| SoftMatch (2023) | 29.96 | 31.84 | 74.38 | 45.39 |
| JointMatch (Ours) | **34.67** | **38.40** | **78.98** | **50.68** |

Table 6: Accuracy results on diverse datasets with a larger number of classes. JointMatch consistently outperforms other methods, showing that our approach also works well in settings with a larger number of classes.

ing. $N_c$ is set to 10 for both the AG News and IMDB datasets, and 20 for the Yahoo! dataset.

| Datasets | # Classes | Total | Train | Val | Test |
|---|---|---|---|---|---|
| Empathetic Dialogues | 32 | 24,850 | 19,533 | 2,770 | 2,547 |
| GoEmotions | 27 | 29,425 | 23,485 | 2,956 | 2,984 |
| Hurricane | 8 | 12,800 | 10,240 | 1,280 | 1,280 |

Table 7: Statistics and split information of the added datasets. All numbers shown here are the total number of data in all classes.

The results are averaged over 5 runs with different model parameter initialization. We observe that our JointMatch consistently demonstrates the best performances across all 3 datasets, surpassing the best baseline by 5.13% on average. On the most challenging Yahoo! Answer dataset, JointMatch also significantly outperforms the best baseline by 5.25% accuracy and 5.13% macro-F1. These substantial improvements indicate the effectiveness of our diverse and collaborative pseudo-labeling framework JoinMatch. We further provide detailed ablation studies and analysis in the next section.

### 3.3 Varying the Number of Labeled Data

We also conduct experiments with varying numbers of labeled data to demonstrate the accuracy improvement of our JointMatch over FixMatch on Yahoo! Answers and AG News datasets. To ensure a fair comparison, we also include the performance of the vanilla ensemble of FixMatch, where we train two FixMatch models with different initializations and average their predicted probability distributions to obtain the result for the ensemble model. As shown in Table 4 and Table 5, JointMatch outperforms the vanilla ensemble of FixMatch with different numbers of labeled data on both datasets. This indicates that our improvement does not come simply from the model ensemble. Noteworthy is that, JointMatch offers remarkable improvements over this vanilla ensemble model, especially with *extremely limited labeled data*: 7% on Yahoo! Answer with 5 labels per class and surprisingly 13.99% on AG News with 5 labels per class. This further validates the effectiveness of JointMatch in utilizing unlabeled data and demonstrates its capability and potential to be used in real-world scenarios.

### 3.4 Generalizability Results

To show the generalizability of JointMatch, we add experiments on three datasets with a larger number of classes (c=32, 27, 8). Table 6 summarizes the accuracy results in the 10-shot setting. We also

| Method | AG News | Yahoo! |
|---|---|---|
| JointMatch | **88.39** | **68.32** |
| – Adaptive Threshold | 82.97 | 66.42 |
| – Cross Labeling | 82.84 | 65.62 |
| – Disagree Weights | 87.89 | 67.47 |
| – All (FixMatch) | 80.25 | 63.80 |

Table 8: Accuracy after removing different parts of JointMatch on AG News with 10 labels per class, Yahoo! Answer with 30 labels per class.

include one more recently published and competitive semi-supervised learning method SoftMatch (Chen et al., 2023) for comparison. JointMatch consistently delivers improvement over other methods, indicating that our approach also works well in settings with a larger number of classes. Table 7 provides the statistics and split information of the added datasets. We provide short descriptions of these datasets in Appendix C.

## 4 Ablation Study and Analysis

### 4.1 Effectiveness of Each Component

To show the effectiveness of each component in JointMatch, we also measure the accuracy performance of JointMatch after removing different parts in Table 8. We observe that the performance decreases after stripping each component, suggesting that all components in JointMatch contribute to the final performance. The performance of JointMatch drops most significantly after removing cross-labeling on both datasets, justifying the benefits of two models teaching each other and its help in alleviating error accumulation. Adaptive local thresholding performs a similarly vital role in the final performance of JointMatch. This indicates the effectiveness of adjusting local thresholds based on current classwise learning status.

Removing *disagree weights*, i.e., weighted disagreement and agreement updates, also gives a performance drop, although smaller compared to the other two components. We hypothesize that the reason for this is that the current fixed disagreement weights are not optimal. They should be adaptively adjusted based on the degree of disagreement between the two networks at different times. This topic is outside the focus and scope of this paper and we leave it for future exploration. Despite this, we show the benefit of disagreement weights and provide an ablation study on its values in the next section.

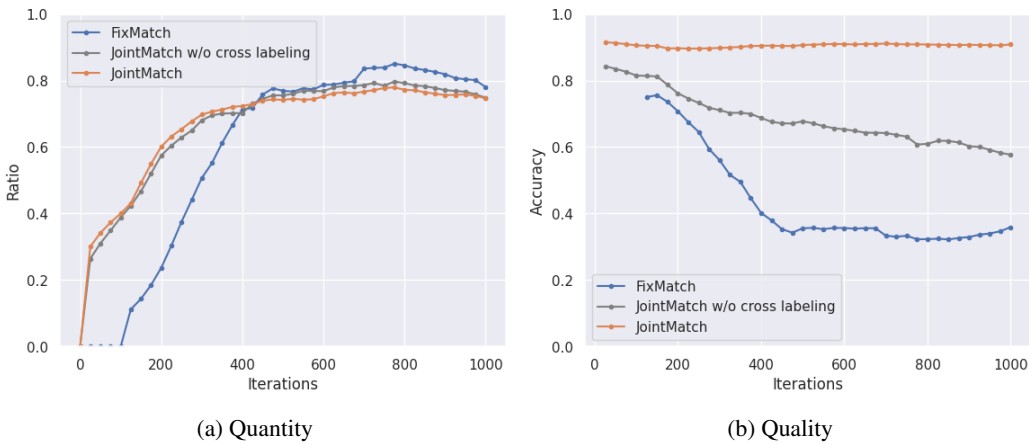

|            | (a) Quantity | (b) Quality |
|------------|--------------|-------------|

Figure 3: The quantity and quality of pseudo labels on AG News with 10 labels per class. Ratio refers to the fraction of pseudo-labeled data that participate in the current training iteration. JointMatch improves the quantity of pseudo labels at the early training stage, while maintaining a high quality of pseudo labels throughout the whole training process.

| $\delta$ | 0 | 0.3 | 0.5 | 0.7 | 0.9 | 1 |
|----------|-------|-------|-------|-------|-------|-------|
| Accuracy | 87.22 | 87.55 | 87.89 | 88.22 | **88.39** | 87.71 |

Table 9: Accuracy on AG News with 10 labels per class and various disagreement weights. Updating networks by only agreement samples ($\delta = 0$) and only disagreement samples ($\delta = 1$) perform worse than updating by both types of data. Weighting more disagreement samples further delivers better results.

## 4.2 Influence of Disagreement Weights

We now analyze the benefits of the weighted disagreement & agreement update. Table 9 shows how our approach performs with different disagreement weights $\delta$ on AG News with 10 labels per class. Note that (1) $\delta = 0$ means updating networks by only agreement samples, $\delta = 1$ means updating by only disagreement samples and $\delta = 0.5$ means updating by both types of samples and weighting them equally; (2) All samples used for updating networks are firstly high confidence samples.

We observe that updating by either only agreement samples or only disagreement samples performs worse than updating by both kinds of samples. This validates our assumption both agreement samples and disagreement samples are beneficial for training, as agreement samples provide high-quality pseudo-labels while disagreement samples offer sample diversity and keep the two networks diverse. In addition, weighting more disagreement samples can obtain better results than equal weighting, which emphasizes the importance of keeping the two networks diverged and thus enabling them to learn from each other.

## 4.3 Quantity & Quality of Pseudo Labels

**JointMatch improves both the quality and quantity of pseudo labels used for training.** As shown in Figure 3a, FixMatch gradually leverages more pseudo labels as training advances, ultimately utilizing about 80% of the available pseudo labels. However, the cost is that the accuracy of pseudo labels keeps dropping, eventually falling below 40%, as indicated in Figure 3b. This is consistent with our motivation in Section 2.3 that directly using noisy pseudo labels will cause models to accumulate errors and gradually produce even more noisy pseudo labels. On the contrary, JointMatch maintains a high accuracy of pseudo labels throughout the entire training process (Figure 3b), which validates the effectiveness of cross-labeling in preventing error accumulation. Furthermore, JoinMatch generates a greater number of pseudo labels than FixMatch during the early stages of training (Figure 3a) without sacrificing their quality.

## 5 Related Work

**Semi-Supervised Text Classification.** Semi-supervised learning has attracted a lot of attention in the field of text classification due to its ability to leverage large amounts of unlabeled data with limited labels. UDA (Xie et al., 2020) introduces strong data augmentation and proposes a consistency loss to minimize the distance of predicted distributions between differently perturbed data. MixText (Chen et al., 2020) leverages Mixup

(Zhang et al., 2018) to interpolate unlabeled data and labeled data in hidden space to avoid overfitting the limited labeled data. S2TC-BDD (Li et al., 2021) notices the margin bias issue and addresses it by balancing label angle variances. PCM (Xu et al., 2022) exploits the inherent semantic matching capability inside pre-trained language models to benefit SSTC. AUM-ST (Sosea and Caragea, 2022) builds on self-training and uses Area Under the Margin (Pleiss et al., 2020) to filter possibly inaccurate pseudo-labels. Hosseini and Caragea (2023) combine two models in a co-training fashion, one being trained using unsupervised domain adaptation from a source to a target domain and the other using semi-supervised learning in the target domain where the two models iteratively teach each other by interchanging their high confident predictions. CrisisMatch (Zou et al., 2023) studies several popular semi-supervised components and integrates Mixup with pseudo-labeling (Lee et al., 2013) for disaster tweet classification. Building upon the idea of FixMatch (Sohn et al., 2020) that uses pseudo-labels generated by weakly-augmented data to teach strongly-augmented data, the recent work SAT (Chen et al., 2022) proposes to re-rank different weak and strong augmentations according to their similarity with the original input. However, these works ignore the difficulties of learning different classes at different time steps. Inspired by recent FlexMatch (Zhang et al., 2021) and FreeMatch (Wang et al., 2023b), our Joint-Match introduces adaptive thresholding to SSTC to dynamically adjust classwise thresholds based on the learning status of each class.

**Learning with Noise.** The task of learning with noise aims to train robust networks against label noise. Co-Teaching (Han et al., 2018) proposes to simultaneously train two networks and mutually select small-loss instances to teach its peer, thus avoiding direct error accumulation within one network. Nevertheless, as the training goes on, the two networks may converge, reduce to one identical network, and the problem of error accumulation resurfaces. Decoupling (Malach and Shalev-Shwartz, 2017) proposes to update models only by the data receiving different predictions from the two networks. Co-Teaching+ (Yu et al., 2019) observes that this strategy can keep the two networks diverged and further boost the performance of Co-Teaching. While these approaches are originally designed for fully supervised settings, our Joint-

Match integrates their ideas into semi-supervised text classification. We provide a comprehensive comparison between closely related approaches with our JointMatch in Table 1.

# 6 Conclusion

In this paper, we propose JointMatch, a semi-supervised text classification approach that integrates ideas from recent semi-supervised learning and learning with noise. Our method is motivated by observed problems in semi-supervised text classification (SSTC): model bias towards easy classes and error accumulation. JointMatch utilizes adaptive thresholding, cross-labeling, and weighted disagreement & agreement updates to address these issues effectively. Through extensive experiments on three standard SSTC benchmark datasets, we found that JointMatch significantly outperforms previous works on all datasets and demonstrates surprising improvement over FixMatch on an extremely scarce label setting. We also show that JointMatch can generalize well to datasets that have a large number of classes.

# Limitations

There are several limitations to our work. First, our method trains two networks simultaneously, resulting in a higher computational cost and longer training time. For example, each experiment on AG News takes approximately 2.5 hours on an A5000 GPU for JointMatch, while FixMatch only requires 1.25 hours. However, the additional computational cost is not a significant issue in low-resource settings, as experiments in such settings typically do not take much time. Second, our weighted disagreement & agreement update could be further improved by adaptively adjusting the disagreement weight based on the degree of mutual agreement among networks and their confidence at different times. This will be explored in the future.

# Acknowledgements

This research is partially supported by National Science Foundation (NSF) grants IIS-1912887, IIS-2107487, and ITE-2137846. Any opinions, findings, and conclusions expressed here are those of the authors and do not necessarily reflect the views of NSF. We thank our reviewers for their insightful feedback and comments which helped improve the quality of our paper.

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

## A  Hyperparameters

A complete list of JointMatch hyperparameters on all evaluated datasets are provided in Table 10.

| | AG News | Yahoo! Answer | IMDB |
|---|---|---|---|
| Batch Size $B$ | 8 | 4 | 4 |
| Learning Rate $lr$ | 1e-5 | 2e-5 | 2e-5 |
| Unsupervised Loss Weight $w_u$ | 1 | 1 | 0.05 |
| EMA decay $\lambda$ | | 0.9 | |
| Fixed Threshold $\tau$ | | 0.98 | |
| Disagreement Weight $\delta$ | | 0.9 | |
| Unlabeled Data Ratio $\mu$ | | 10 | |

Table 10: Complete list of JointMatch hyperparameters on AG News, Yahoo! Answer, IMDB.

## B  Additional Ablation Study Results

This section provides additional ablation study results on several hyperparameters as complementary to the main paper. All experimental results here are obtained from JointMatch on AG News with 10 labels per class.

### B.1  EMA Decay

Table 11 shows the result of JointMatch with different EMA decay $\lambda$ for estimating classwise learning status. Note that $\lambda = 1$ means the estimated learning status for all classes always equals their initial value $1/C$ and not adjusting local thresholds ($C$ is the number of classes). We observe that adjusting local thresholds based on estimated learning ($\lambda \in [0, 1)$) is significantly better than not adjusting local thresholds ($\lambda = 1$).

| EMA Decay | Accuracy | Macro-F1 |
|---|---|---|
| 0 | 87.13 | 87.06 |
| 0.25 | 86.93 | 86.95 |
| 0.5 | 87.61 | 87.59 |
| 0.9 | **88.39** | **88.32** |
| 0.99 | 87.28 | 87.28 |
| 1 | 83.68 | 83.8 |

Table 11: Ablation study on EMA decay.

### B.2  Fixed Confidence Threshold

In Table 12, we show the performance of Joint-Match with varying fixed confidence threshold $\tau$. It can be seen that setting a threshold higher than 0.75 is generally beneficial for good performance, but setting it to a too high value will lead to some

| Fixed Threshold | Accuracy | Macro-F1 |
|---|---|---|
| 0 | 83.13 | 83.32 |
| 0.25 | 83.13 | 83.32 |
| 0.5 | 85.21 | 85.13 |
| 0.75 | 87.32 | 87.37 |
| 0.9 | 87.47 | 87.49 |
| 0.95 | 87.37 | 87.3 |
| 0.98 | **88.39** | **88.32** |
| 0.99 | 87.84 | 87.82 |

Table 12: Ablation study on the fixed threshold.

performance drop, as less number of unlabeled can be utilized.

### B.3  Unlabeled Data Ratio in MiniBatch

In Table 13, we present the results of JointMatch with different unlabeled data to labeled data ratio $\mu$. One can observe that using a large amount of unlabeled data can help increase performance. A very high value of $\mu$ is not encouraged since it slows down the training while just giving marginal improvements.

| Unlabeled Data Ratio | Accuracy | Macro-F1 |
|---|---|---|
| 1 | 86.43 | 86.39 |
| 3 | 86.74 | 86.71 |
| 5 | 88.08 | 88.04 |
| 10 | 88.39 | 88.32 |
| 15 | 88.22 | 88.22 |
| 20 | 88.45 | 88.46 |
| 30 | 88.49 | 88.50 |

Table 13: Ablation Study on Unlabeled Data to Labeled Data Ratio.

## C  Short Descriptions of Added Datasets

We provide a short description of these datasets here: (1) GoEmotions (Demszky et al., 2020) is a dataset of Reddit comments labeled with 27 emotions, such as amusement, fear, and gratitude. (2) Empathetic Dialogues (Rashkin et al., 2019) consists of conversations between a speaker and listener and is labeled with 32 fine-grained emotions. (3) Hurricane is a crisis tweet dataset sampled from HumAID (Alam et al., 2021). It contains human-labeled tweets collected during hurricane disasters and includes 8 crisis-related classes, such as infrastructure and utility damage, displaced people and evacuations.