# OpenReview forum: "JointMatch: A Unified Approach for Diverse and Collaborative Pseudo-Labeling to Semi-Supervised Text Classification"
_EMNLP/2023/Conference — EMNLP 2023 Main_

### Official Review · Reviewer_zxwi · 2023-07-20

**Soundness:** 3

**Excitement:**

2: Mediocre: This paper makes marginal contributions (vs non-contemporaneous work), so I would rather not see it in the conference.

**Paper Topic And Main Contributions:**

The paper deals with the topic of Semi-supervised Text Classification (SSTC), an area that has been receiving increasing attention because of its ability to leverage unlabeled data. The central issue that the paper addresses is the problems of pseudo-label bias and error accumulation in SSTC, which are common issues in existing approaches that use pseudo-labeling.

The main contribution of the paper is the proposal of a new approach, JointMatch, which unifies ideas from recent semi-supervised learning and learning with noise. The approach adjusts class-wise thresholds based on the learning status of different classes to mitigate model bias towards currently easy classes. It also alleviates error accumulation by utilizing two differently initialized networks to teach each other in a cross-labeling manner.

Additionally, JointMatch introduces a strategy that weights more disagreement data while also allowing the utilization of high-quality agreement data for training. The paper presents experimental results on benchmark datasets that demonstrate the superior performance of JointMatch, achieving a significant improvement. Remarkably, JointMatch delivers impressive results even in the extremely scarce label setting.

**Questions For The Authors:**

Please add more experiments and I will reconsider my score.

**Reasons To Accept:**

1. The paper addresses an important issue in semi-supervised text classification, i.e., pseudo-label bias and error accumulation.
2. The JointMatch approach is innovative, combining recent ideas from semi-supervised learning and learning with noise.
3. The paper provides experimental evidence of the effectiveness of JointMatch, including in scenarios with very few labels.
4. The approach introduced in the paper is robust and can handle disagreement data and high-quality agreement data effectively.

**Reasons To Reject:**

1. The complexity of the JointMatch approach, which might limit its applicability or use in certain scenarios.
2. It would be interesting to see how JointMatch performs compared to other semi-supervised learning methods, not just the baseline methods. A broader comparison could have been beneficial.
3. Depending on the experimental results and discussions, there could be limitations in the evaluation of the approach. For instance, the performance of the approach might not be tested across a sufficiently diverse range of datasets or problem domains.

**Reproducibility:**

4: Could mostly reproduce the results, but there may be some variation because of sample variance or minor variations in their interpretation of the protocol or method.

**Reviewer Confidence:**

5: Positive that my evaluation is correct. I read the paper very carefully and I am very familiar with related work.

---

> ### Author Rebuttal · Authors · 2023-08-29
>
> Many thanks for your valuable comments and assessment of our paper!
>
> > Q1: It would be interesting to see how JointMatch performs compared to other semi-supervised learning methods, not just the baseline methods. A broader comparison could have been beneficial.
>
> > Q2: Depending on the experimental results and discussions, there could be limitations in the evaluation of the approach. For instance, the performance of the approach might not be tested across a sufficiently diverse range of datasets or problem domains. Please add more experiments and I will reconsider my score.
>
> Thank you for these constructive suggestions. We have **added more** recent semi-supervised methods and diverse datasets. The table below compares the accuracy of JointMatch with **recently published and competitive semi-supervised learning methods**: SoftMatch (Chen et al., 2023 ICLR), FreeMatch (Wang et al., 2023 ICLR), SAT (Chen et al., 2022 EMNLP) and baseline FixMatch (Sohn et al., 2020 NeurIPS). We conducted experiments on **six different datasets**, covering a **diverse** range of problem **domains**. The results show that JointMatch consistently outperforms these powerful semi-supervised learning methods on diverse datasets and tasks. A more detailed table will be added to our paper. Furthermore, we provide a short description of these datasets below. If you are interested in any other specific methods, datasets or experiments, please let us know, and we will be happy to conduct experiments immediately and include them in our paper. Thanks again for being willing to reconsider your scores.
>
>
> | Dataset           | Empathetic Dialogues | GoEmotions | Yahoo! Answers | Hurricane | AG News | IMDB   | Total Average |
> | ----------------- | -------------------- | ---------- | -------------- | --------- | ------- | ------ | ------------- |
> | \# Classes        | 32                   | 27         | 10             | 8         | 4       | 2      | \-            |
> | \# Total Data     | 24,850               | 29,425     | 130,000        | 12,800    | 35,600  | 37,000 | \-            |
> | BERT (2019)       | 21.63                | 28.86      | 58.11          | 71.72     | 69.18   | 63.16  | 52.11         |
> | FixMatch (2020)   | 24.4                 | 30.06      | 60.17          | 73.44     | 80.22   | 64.52  | 55.47         |
> | SAT (2022)        | 29.25                | 30.36      | 61.51          | 74.06     | 86.38   | 65.43  | 57.83         |
> | SoftMatch (2023)  | 29.96                | 31.84      | 64.83          | 74.38     | 82.68   | 67.92  | 58.60         |
> | FreeMatch (2023)  | 30.15                | 32.74      | 62.11          | 77.03     | 85.96   | 70.10  | 59.68         |
> | JointMatch (Ours) | **34.67**                | **38.4**       | **66.58**          | **78.98**     | **87.68**   | **76.91**  | **63.87**         |
>
>
> > Q3: The complexity of the JointMatch approach, which might limit its applicability or use in certain scenarios.
>
> We acknowledge the complexity of JointMatch since it unifies ideas from different domains to address two critical issues in semi-supervised text classification. However, it yields significant improvements in diverse datasets and settings. If the user does not want to deal with the full complexity of the model, they can remove some modules, such as disagreement and agreement weights. The model will still perform well ( though not as well as the full model), as shown in the ablation study from Table 5 in the paper. Besides, we provide a legible algorithm of JointMatch in the paper and release our code to help users understand and utilize our method.
>
> ---
>
> **Short Descriptions of Datasets:** [1] GoEmotions (Demszky et al., 2020) is a dataset of Reddit comments labeled with 27 emotions, such as amusement, fear, and gratitude. [2] Empathetic Dialogues (Rashkin et al., 2019) consists of conversations between a speaker and listener and is labeled with 32 fine-grained emotions. [3] Hurricane is a crisis tweet dataset sampled from HumAID (Alam et al., 2021). It contains human-labeled tweets collected during hurricane disasters and includes 8 crisis-related classes, such as infrastructure_and_utility_damage, displaced_people_and_evacuations. Next are the datasets that are already in the paper: [4] Yahoo! Answers classifies question-answer pairs into 10 topics, such as Sports, Health, and Education. [5] AG News categorizes news articles into 4 categories, such as World, Sci/Tech, and Business. [6] IMDB is a binary sentiment analysis dataset consisting of movie reviews. We will provide a more comprehensive description of datasets, statistics, and split information in our paper.
>
> [1] Dorottya Demszky, Dana Movshovitz-Attias, Jeongwoo Ko, Alan Cowen, Gaurav Nemade, and Sujith Ravi. 2020. GoEmotions: A dataset of fine-grained emotions. In Proceedings of the 58th Annual Meeting of the Association for Computational Linguistics, pages 4040–4054, Online. Association for Computational Linguistics.
>
> [2] Hannah Rashkin, Eric Michael Smith, Margaret Li, and Y-Lan Boureau. 2019. Towards empathetic open domain conversation models: A new benchmark and dataset. In Proceedings of the 57th Annual Meeting of the Association for Computational Linguistics, pages 5370–5381, Florence, Italy. Association for Computational Linguistics.
>
> [3] Firoj Alam, Umair Qazi, Muhammad Imran, and Ferda Ofli. 2021. Humaid: human-annotated disaster incidents data from twitter with deep learning benchmarks. In Proceedings of the International AAAI Conference on Web and Social Media, volume 15, pages 933–942.
>
> [4] Ming-Wei Chang, Lev Ratinov, Dan Roth, and Vivek Srikumar. 2008. Importance of semantic representation: Dataless classification. In Proceedings of the 23rd National Conference on Artificial Intelligence - Volume 2, AAAI’08, page 830–835. AAAI Press.
>
> [5] Xiang Zhang, Junbo Zhao, and Yann LeCun. 2015. Character-level convolutional networks for text classification. Advances in neural information processing systems, 28.
>
> [6] Andrew L. Maas, Raymond E. Daly, Peter T. Pham, Dan Huang, Andrew Y. Ng, and Christopher Potts. 2011. Learning word vectors for sentiment analysis. In Proceedings of the 49th Annual Meeting of the Association for Computational Linguistics: Human Language Technologies, pages 142–150, Portland, Oregon, USA. Association for Computational Linguistics.

---

### Official Review · Reviewer_Tckf · 2023-07-28

**Soundness:** 4

**Excitement:**

4: Strong: This paper deepens the understanding of some phenomenon or lowers the barriers to an existing research direction.

**Missing References:**

-

**Paper Topic And Main Contributions:**

The paper introduces a new approach called JointMatch for semi-supervided text classification. It follows the idea of pseudo-labeling while avoiding the well-known problem of bias and error accumulation in simple pseudo-labeling approaches with a fixed threshold.It gives a good overview of how the new approach can be compared to closely related methods (table 1).

**Questions For The Authors:**

Would it be possible to apply the approach also in a dynamic setting, e.g. when new topics are showing up?

**Reasons To Accept:**

The paper is well written. It contains most of the relevant information to follow the ideas and method setup. In the experimental study the approach shows better performance than the other methods with which it is compared. The method seems to work well even with a very small number on labeled cases per class.

**Reasons To Reject:**

One major drawback of the paper is in my opinion that the number of classes of the datasets included in the experiment are relatively small (c=4, c=10, c=2). It has to be proven that the approach also works well in settings with a large number of classes.

Concerning the performance: the standard deviation (SD) seems to be rather high for the IMDB dataset. In general: if we take e.g. for the accuracy the "average +/- 2 times the SD", the proposed method several times overlaps with the SAT method. So the performance improvements look less impressive from this point of view.

The details on the data augmentation are not provided.

**Reproducibility:**

4: Could mostly reproduce the results, but there may be some variation because of sample variance or minor variations in their interpretation of the protocol or method.

**Reviewer Confidence:**

3: Pretty sure, but there's a chance I missed something. Although I have a good feel for this area in general, I did not carefully check the paper's details, e.g., the math, experimental design, or novelty.

**Typos Grammar Style And Presentation Improvements:**

-

---

> ### Author Rebuttal · Authors · 2023-08-29
>
> Many thanks for your valuable comments and assessment of our paper!
>
> > Q1: One major drawback of the paper is in my opinion that the number of classes of the datasets included in the experiment are relatively small (c=4, c=10, c=2). It has to be proven that the approach also works well in settings with a large number of classes.
>
> Thank you for this constructive suggestion. We have **added** experiments on **three datasets** with a larger number of classes (c=32, 27, 8), and the accuracy results are shown in the table below. JointMatch consistently delivers improvement over other methods, showing that our approach also **works well in settings with a larger number of classes**. We provide short descriptions of the added datasets below. Please let us know if you are interested in any other specific datasets, we will be happy to conduct experiments immediately and add results on them.
>
> | Dataset           | Empathetic | Dialogues |  (c=32) |       | GoEmotions | (c=27) |       |       | Hurricane  | (c=8) |       |       | Average |
> |-----------------|:----------:|:---------:|:-------:|:-----:|:----------:|:------:|:-----:|:-----:|:----------:|:-----:|:-----:|:-----:|:-------:|
> | #Labels/Class     | 5          | 10        | 15      | 25    | 5          | 10     | 15    | 25    | 5          | 10    | 15    | 25    | -       |
> | BERT              | 15.54      | 21.63     | 29.25   | 33.80 | 18.26      | 28.86  | 37.86 | 44.33 | 24.92      | 53.98 | 66.25 | 71.72 | 37.20   |
> | FixMatch          | 16.53      | 24.40     | 34.75   | 37.81 | 24.60      | 30.06  | 34.65 | 42.06 | 34.30      | 51.09 | 65.62 | 73.44 | 39.11   |
> | FreeMatch         | 17.86      | 30.15     | 36.08   | 39.14 | 24.50      | 32.74  | 35.76 | 44.34 | 35.78      | 52.03 | 67.50 | 77.03 | 41.08   |
> | FixMatch-Ensemble | 20.22      | 30.31     | 37.50   | 42.25 | 27.45      | 36.16  | 39.21 | 45.04 | 35.23      | 56.48 | 67.50 | 73.67 | 42.59   |
> | JointMatch        | **22.69**      | **34.67**     | **40.40**   | **44.80** | **30.97**      | **38.40**  | **40.11** | **45.64** | **45.23**      | **73.59** | **76.41** | **78.98** | **47.66**   |
>
>
>
>
> > Q2: Concerning the performance: the standard deviation (SD) seems to be rather high for the IMDB dataset. In general: if we take e.g. for the accuracy the "average +/- 2 times the SD", the proposed method several times overlaps with the SAT method. So the performance improvements look less impressive from this point of view.
>
> We conducted the **statistical t-test** to show the significant difference between the accuracy result of JointMatch (76.91 ± 4.5) and SAT (68.96 ± 1.7) on IMDB dataset. The computed t-value = 2.698 is more extreme than the critical t-value 2.306 ($\alpha = 0.05$). This test shows that **JointMatch is significantly better than SAT on IMDB dataset**. Besides, to further address the concern about the performance, we report the average accuracy results across 6 diverse datasets in the table below. We can see that the total average performance **improvement** of JointMatch **is significant across 7 diverse datasets**.
>
>
> | Dataset           | Empathetic Dialogues | GoEmotions | Yahoo! Answers | Hurricane | AG News | IMDB   | Total Average |
> | ----------------- | -------------------- | ---------- | -------------- | --------- | ------- | ------ | ------------- |
> | \# Classes        | 32                   | 27         | 10             | 8         | 4       | 2      | \-            |
> | \# Total Data     | 24,850               | 29,425     | 130,000        | 12,800    | 35,600  | 37,000 | \-            |
> | BERT (2019)       | 21.63                | 28.86      | 58.11          | 71.72     | 69.18   | 63.16  | 52.11         |
> | FixMatch (2020)   | 24.4                 | 30.06      | 60.17          | 73.44     | 80.22   | 64.52  | 55.47         |
> | SAT (2022)        | 29.25                | 30.36      | 61.51          | 74.06     | 86.38   | 65.43  | 57.83         |
> | SoftMatch (2023)  | 29.96                | 31.84      | 64.83          | 74.38     | 82.68   | 67.92  | 58.60         |
> | FreeMatch (2023)  | 30.15                | 32.74      | 62.11          | 77.03     | 85.96   | 70.10  | 59.68         |
> | JointMatch (Ours) | **34.67**                | **38.4**       | **66.58**          | **78.98**     | **87.68**   | **76.91**  | **63.87**         |
>
>
> > Q3: The details on the data augmentation are not provided.
>
> Thanks for bringing this to our attention. Following previous work SAT (Chen et al., 2022), we use back translation as strong data augmentation and employ synonym replacement as weak data augmentation. In detail, for back translation, we translate texts into German and then translate them back into English; for synonym replacement, we randomly substitute 30% of words with WordNet synonyms. The code of our augmentation techniques will be provided.
>
> > Q4: Would it be possible to apply the approach also in a dynamic setting, e.g. when new topics are showing up?
>
> Thank you for pointing it out. This is an interesting direction and deserves another study on different scenarios: specifically, when new topics arise, we may have different scenarios, for example, (1) limited labeled data and limited unlabeled data, or (2) limited labeled data along with large amounts of unlabeled data. Given our encouraging results on the extremely low-shot setting, we believe our method has the potential to be effective in these scenarios. We will explore these in the future and will include a discussion on this in our final version of the paper.
>
> We sincerely hope our response can resolve your concerns. But if you have any further concerns or questions, please let us know, and we will do our best to address them as soon as we can.
>
> ---
> **Short Descriptions of the Added Datasets**: [1] GoEmotions (Demszky et al., 2020) is a dataset of Reddit comments labeled with 27 emotions, such as amusement, fear, and gratitude. [2] Empathetic Dialogues (Rashkin et al., 2019) consists of conversations between a speaker and listener and is labeled with 32 fine-grained emotions. [3] Hurricane is a crisis tweet dataset sampled from HumAID (Alam et al., 2021). It contains human-labeled tweets collected during hurricane disasters and includes 8 crisis-related classes, such as infrastructure_and_utility_damage, displaced_people_and_evacuations. We will provide a more comprehensive description of datasets, statistics and split information in our paper.
>
> [1] Dorottya Demszky, Dana Movshovitz-Attias, Jeongwoo Ko, Alan Cowen, Gaurav Nemade, and Sujith Ravi. 2020. GoEmotions: A dataset of fine-grained emotions. In Proceedings of the 58th Annual Meeting of the Association for Computational Linguistics, pages 4040–4054, Online. Association for Computational Linguistics.
>
> [2] Hannah Rashkin, Eric Michael Smith, Margaret Li, and Y-Lan Boureau. 2019. Towards empathetic open domain conversation models: A new benchmark and dataset. In Proceedings of the 57th Annual Meeting of the Association for Computational Linguistics, pages 5370–5381, Florence, Italy. Association for Computational Linguistics.
>
> [3] Firoj Alam, Umair Qazi, Muhammad Imran, and Ferda Ofli. 2021. Humaid: human-annotated disaster incidents data from twitter with deep learning benchmarks. In Proceedings of the International AAAI Conference on Web and Social Media, volume 15, pages 933–942.

---

### Official Review · Reviewer_3ZiN · 2023-08-03

**Soundness:** 4

**Excitement:**

4: Strong: This paper deepens the understanding of some phenomenon or lowers the barriers to an existing research direction.

**Paper Topic And Main Contributions:**

In this paper, the authors consider the semi-supervised text classification setting which has gained a lot of interest due to its promise of enhancing classification performance with limited labeled data. Specifically, they focus on collaborative pseudo labelling and consistency regularization citing two limitations of existing approaches - (1) fixed threshold for pseudo labelling leading to biased label generation for easy to learn classes; (2) progressive accumulation of pseudo labelling errors leading to eventual degradation of classification performance.
They then propose an approach that they refer to as JointMatch that makes use of adaptive per class thresholding to address the biased label generation and cross-labelling with two classification models trained on divergence/consensus weighted loss. Adaptive local thresholding adjusts the per class threshold based on the learning status of the class and progressively updates it to prevent starvation of hard-to-learn classes. Cross-labelling makes use of two different classification networks that are separately initialized and leverages the thesis that different networks filter out different noises. This helps in alleviating the accumulation of errors. However, co-training, the driving idea behind cross-labelling, eventually converges and degenerates to self-training. In order to retain the divergence of these networks, the authors train them on an objective that is weighted by the divergence / convergence of these networks.
Evaluation on multiple datasets across different baseline models and ablation study supports the efficacy of the proposed approach.

**Questions For The Authors:**

Refer above.

**Reasons To Accept:**

While deep learning has become ubiquitous, training these models often requires large amounts of data that has cost and environmental impact. Data efficient learning, including semi-supervised learning that this paper deals with, is an important research direction. The authors highlight two important challenges with the current semi-supervised classification approaches and propose ways to address these. The proposed techniques, although inspired from existing work, are gracefully combined into a holistic approach.
They also make the code available which will help practitioners leverage this technique. Researchers can replicate the work and potentially extend it further.
The approach is well described and the paper is generally well written.
The evaluation is thorough and offers sufficient support to the claims around addressing the challenges highlighted earlier in the paper.

**Reasons To Reject:**

All the three techniques of adaptive local thresholding, cross-labelling and weighted disagreement / agreement update are based on existing work. In that sense, the paper is weak on the novelty aspect.
There are some hypotheses that are not backed by sufficient analytical explanation. For instance, "different networks can filter out different noises" - it is unclear why two networks are enough to guarantee this? What if both the networks tend to make similar errors?
The pseudo labels generated for classes with low confidence threshold - won't these have higher probability of being incorrect?

While the evaluation suggests that this works well in practise, the paper can benefit from a more sound theoretical backing.

**Reproducibility:**

4: Could mostly reproduce the results, but there may be some variation because of sample variance or minor variations in their interpretation of the protocol or method.

**Reviewer Confidence:**

4: Quite sure. I tried to check the important points carefully. It's unlikely, though conceivable, that I missed something that should affect my ratings.

**Typos Grammar Style And Presentation Improvements:**

Equation 2 --> missing bracket
"local threshold" is introduced in section 2.1 but the notations for it come later in section 2.2

---

> ### Author Rebuttal · Authors · 2023-08-29
>
> Many thanks for your valuable comments and assessment of our paper!
>
> > Q1: All the three techniques of adaptive local thresholding, cross-labeling and weighted disagreement/agreement update are based on existing work. In that sense, the paper is weak on the novelty aspect.
>
> We agree that our work combines techniques from existing works, but so do other works, e.g., FixMatch which combines consistency regularization and pseudo-labeling. However, our novelty exhibits in many aspects: (1) Our JointMatch is innovative at proposing a holistic framework that effectively unifies ideas from recent semi-supervised learning and learning with noise for semi-supervised text classification, which is also pointed out by Reviewer zxwi. The relation and difference with related approaches are also summarized in Table 1 and Section 2.5. (2) Previous works Decoupling and Co-Teaching+ completely ignore agreement data when updating their network. We argue that those agreement data are valuable learning signals and propose a new strategy that utilizes both disagreement and agreement data in a weighted manner. (3) To the best of our knowledge, our work is the **first in semi-supervised text classification** that **highlights** and **effectively addresses both** the **issues** of **error accumulation** and **bias toward easy classes** in **one unified framework**, by combining cross-labeling, weighted agreement update and adaptive thresholds.
>
> > Q2: There are some hypotheses that are not backed by sufficient analytical explanation. For instance, "different networks can filter out different noises" - it is unclear why two networks are enough to guarantee this? What if both networks tend to make similar errors?
>
> Thank you for giving us the opportunity to clarify. We present statistics about the filtered *incorrect pseudo-label* (noise) of every 50 iterations and 4K predictions in the table below. (1) We can observe that there is a considerable amount/percentage of non-overlapping noises between two networks in different iterations, validating that different networks can filter out different noises. This hypothesis is also supported by previous work Co-Training (Blum & Mitchell, 1998) and Co-Teaching (Han et al., 2018b). (2) For those overlapping noises, i.e., the same errors made by both networks, we can see that models will reduce their percentages and improve upon them over iterations.
>
>
> | Iteration | \# Non-Overlap Noise | \# Overlap Noise | Non-Overlap Percentage | Overlap Percentage |
> | --------- | -------------------- | ---------------- | ---------------------- | ------------------ |
> | 50        | 594                  | 1328             | 0.145                  | 0.335              |
> | 100       | 502                  | 1044             | 0.125                  | 0.260              |
> | 150       | 509                  | 890              | 0.130                  | 0.220              |
> | 200       | 401                  | 848              | 0.100                  | 0.210              |
> | 250       | 380                  | 905              | 0.095                  | 0.225              |
> | 300       | 364                  | 819              | 0.095                  | 0.205              |
> | 350       | 281                  | 776              | 0.075                  | 0.190              |
>
>
> > Q3: The pseudo labels generated for classes with low confidence threshold - won't these have higher probabilities of being incorrect?
>
> It is correct that pseudo-labels generated from lower confidence thresholds are more likely to be incorrect. (1) However, adaptively lowering confidential thresholds for hard classes allows for pseudo-label diversity and can help alleviate aggressive bias towards easy classes. (2) Our method is designed to deal with the case where the pseudo-labels are potentially incorrect. First, as the training goes on, the model will improve its accuracy in these hard classes and progressively increase their confidence thresholds to filter out low-quality pseudo-labels. Second, double-networks, cross-labeling and weighted-agreement-update can effectively maintain a high quality of pseudo-labels throughout the whole training process, as visualized in our Figure 3.
>
> Hope our response could resolve your concern. Please let us know if you have further questions. Thanks again!

---

### Meta-Review · Area_Chair_naPo · 2023-09-19

**Recommendation:** 4

**Metareview:**

The paper proposes JointMatch which addresses the challenges (pseudo-label bias and error accumulation) in semi-supervised text classification (SSTC).

JointMatch combines three existing techniques (cross-labeling, per-class adaptive thresholding, and weighted (dis) agreement update) in order to mitigate two important problems in SSTC: model bias towards easy-to-learn classes and error accumulation. Experiments on multiple datasets indicate the potential efficacy of JointMatch.

There were worries that the proposed method is only a combination of existing work and not novel for that reason, however, the holistic combination of previous methods and experimental analyses of Jointmatch were recognized as innovative by multiple reviewers. Additionally, the authors added additional results: recent semi-supervised learning models as baselines and experiments of JointMatch on larger label sizes (>10).

I recommend authors reflect new experiment results and clarifications in the manuscript.

---

### Decision · Program_Chairs · 2023-10-07

**Decision:**

Accept-Main

**Comment:**

The paper proposes JointMatch which addresses the challenges (pseudo-label bias and error accumulation) in semi-supervised text classification (SSTC).

JointMatch combines three existing techniques (cross-labeling, per-class adaptive thresholding, and weighted (dis) agreement update) in order to mitigate two important problems in SSTC: model bias towards easy-to-learn classes and error accumulation. Experiments on multiple datasets indicate the potential efficacy of JointMatch.

There were worries that the proposed method is only a combination of existing work and not novel for that reason, however, the holistic combination of previous methods and experimental analyses of Jointmatch were recognized as innovative by multiple reviewers. Additionally, the authors added additional results: recent semi-supervised learning models as baselines and experiments of JointMatch on larger label sizes (>10).

I recommend authors reflect new experiment results and clarifications in the manuscript.